# Antimicrobial Resistance in *Escherichia coli* and Resistance Genes in Coliphages from a Small Animal Clinic and in a Patient Dog with Chronic Urinary Tract Infection

**DOI:** 10.3390/antibiotics9100652

**Published:** 2020-09-29

**Authors:** Veronika Zechner, Dmitrij Sofka, Peter Paulsen, Friederike Hilbert

**Affiliations:** Institute of Food Safety, Department of Farm Animals and Veterinary Public Health, University of Veterinary Medicine, 1210 Vienna, Austria; veronikazechner1977@gmail.com (V.Z.); Dimitrij.Sofka@vetmeduni.ac.at (D.S.); Peter.Paulsen@vetmeduni.ac.at (P.P.)

**Keywords:** *Escherichia coli*, urinary tract infection, transfer, resistance genes

## Abstract

Antimicrobial resistance is on the rise in certain pathogens that infect pets and their owners. This has raised concerns about the use of antibiotics and the transfer of resistance elements in small animal clinics. We sampled a surgery unit, diagnostic rooms after disinfection, and a dog with chronic urinary tract infection (UTI), in a small animal clinic in Austria, and isolated/characterized phages and *Escherichia* (*E.*) *coli* for antimicrobial resistance, resistance genes and transduction ability. Neither the coliphages nor *E. coli* were isolated in the 20 samples of the surgery units and diagnostic rooms. From the urinary tract of the dog, we recovered 57 *E. coli* isolates and 60 coliphages. All of the *E. coli* isolates were determined as resistant against nalidixic acid, 47 against ampicillin, 34 against sulfonamides, and 33 against streptomycin. No isolate held resistance against tetracycline, trimethoprim, kanamycin, or chloramphenicol. Among the 60 phages, 29 tested positive for one or more resistance gene(s) by PCR, but none was able to transduce it to a laboratory strain or to an *E. coli* isolated from samples. Nevertheless, six phages out of 60 were able to transduce ampicillin resistance (*bla* gene) after being grown on a *puc19* harboring *E. coli* strain.

## 1. Introduction

Antimicrobial resistance and antimicrobial usage in small animal veterinary medical clinics are of concern. The shortcomings of antibiotic treatment options and resistance transfer within veterinary clinics is scary, while the transfer of resistant bacteria or resistance elements to the patient’s holders is considered a public health issue. Bacterial resistance develops and is transferred by a variety of mechanisms, such as mutation, transformation, conjugation, and the uptake of naked DNA [1,2,3]. Transduction is one of these mechanisms to transfer resistance genes by bacteriophages viruses that infect bacterial cells. Even though this mechanism does not play a predominant role in antimicrobial resistance transfer, some studies have shown that phage transduction cannot be neglected for the horizontal gene transfer of resistance genes [4,5,6].

Antimicrobial resistance, and the occurrence of multidrug resistant pathogens, such as different *Enterobacteriaceae*, staphylococci, enterococci, *Pseudomonas aeruginosa*, and *Bordetella* spp., cannot be neglected any more in small animal veterinary medicine. Many of these multidrug-resistant pathogens have been described in pets, and have intruded clinical practices, with multidrug-resistant *E. coli* being one of the most important of the kind [7,8]. Dissemination of multidrug resistant isolates to pet owners has been illustrated [9] with negative effects on public health [10]. Identification of carriers is also an important issue. The spread between pet owners, clinical staff, and animals visiting the clinics has to be controlled based on hygiene strategies, relying on scientific evidence and novel results. One of the issues to be considered is how to avoid the transfer of resistant bacteria and resistance genes using adequate disinfection regimes. Recently, we were able to show that phages are important in resistance transfer; some phages withstand certain disinfectants, and that inadequate disinfection may lead to the possibility of resistance spread by phages [11,12].

In this work, we analyzed a small animal veterinary clinic in Austria for *E. coli*, coliphages, and the ability of these phages to transduce antimicrobial resistance. Here, sampling involved sponging the surgery and the diagnostic rooms after disinfection and sampling a dog with a chronic urinary tract infection (UTI) caused by *E. coli*. The phages were characterized by lytic profiles holding antimicrobial resistance genes and by DNA restriction analysis for a selected part of phages. *E. coli* were defined by antimicrobial drug resistance and phage lysis. A possible spread of the phages proceeding from this patient was ruled out, and the functionality of the disinfection regime could be confirmed. 

## 2. Results

### 2.1. Isolation of Coliphage and E. coli from the Clinical Environments

Swabs and sponges from all sampling sites were analyzed for coliphages and *E. coli*. Neither coliphages nor *E. coli* were isolated in the 20 swab and sponge samples of the surgery units and diagnostic rooms.

### 2.2. Antimicrobial Resistance in Isolated E. coli

At this clinic, a dog with chronic urinary tract infection was examined. The dog’s urine was taken at two time points with an interval of 10 days under sterile precautions using a urinary catheter delivering 10^8^
*E. coli* cfu/mL urine. In total, 57 *E. coli* isolates were randomly selected and subjected to antimicrobial resistance testing against ampicillin, chloramphenicol, kanamycin, nalidixic acid, streptomycin, sulfonamides, tetracycline, and trimethoprim. No isolate was susceptible to all of the tested antibiotics. All of the isolates were susceptible to chloramphenicol, kanamycin, and tetracycline. Thirty-five isolates tested resistant to a combination of ampicillin, nalidixic acid, sulfonamides, and streptomycin. Twelve isolates showed resistance against the combination of ampicillin and nalidixic acid; 10 isolates were resistant to nalidixic acid only; and one isolate tested resistant against ampicillin, nalidixic acid, and trimethoprim. The dog was not treated with antibiotics for two years. All of the resistant isolates (out of nalidixic acid-resistant isolates, as resistance to nalidixic acid is, in most cases, caused by mutation) were subjected to DNA extraction and PCR for detection of appropriate resistance genes. In thirty-nine ampicillin-resistant isolates, the *bla*_TEM_ gene fragment was detected; all streptomycin-resistant isolates tested positive for the *strA* gene fragment. In two sulfonamide-resistant isolates, both the *sulI* and *sulII* fragments were identified, whereas in all other sulfonamide-resistant isolates, the *sulII* fragment was detected. In the one trimethoprim-resistant isolate, the *dfr* gene fragment tested positive by PCR (Table 1).

### 2.3. Phages and Antimicrobial Resistance Gene Detection

In total, 60 coliphages were isolated from the dog’s urine, taken, as described, for the isolation of *E. coli*. The phages were selected based on the plaque-forming ability on the reference strain DSM 12242, obtained from the German Culture Collection (DSM). The DNA was cleaned and isolated from these phages and subjected to PCR reaction against antimicrobial resistance genes. Twenty-nine of these phages harbored one of the resistance genes identified in the isolated *E. coli*, namely the *bla*_TEM,_
*sulI*, *sulII*, and/or *strA* gene. In thirteen phages, the *bla*_TEM_ gene fragment was identified. Most often, the *strA* gene fragment was detected—this was the case in 23 of the phage isolates. The sulfonamide-resistant gene fragment *sulI* was only identified in one phage, whereas *sulII* was found in 18 phages. In some phages, more than one resistance gene could be detected by PCR. In eight phages, three of the five tested genes were found by PCR—the *bla*_TEM,_
*strA*, and *sulII* gene combination. In five phages, two of the resistance gene fragments were identified: in two phages the *strA/sulII* gene combination, and in one phage, each, the combination of *bla*_TEM_/*sulI*, *bla*_TEM_/*sulII*, and *bla*_TEM_/*strA*, respectively.

### 2.4. Resistance Transduction

As 29 of the phages harbored one or more resistance gene fragments, we tried to transduce each resistance determinant to the laboratory strain DSM 12242 with different concentrated lysate preparations. Additionally, the transduction was tested in isolated phenotypically nalidixic acid-resistant isolates of *E. coli* from this study (*n* = 10) (as no susceptible *E. coli* was isolated in this study). Nevertheless, transduction of the identified resistance genes was not possible, neither into the laboratory strain DSM 12242 nor into one of the urinary *E. coli* isolates. All 60 phages were used to lyse DH5α-harboring puc19. Out of the 60 phages, 47 were able to lyse DH5α-harboring puc19 and six of these were able to transduce the *bla* gene from puc19 into DSM 12242 without concurrently transferring the intact plasmid.

### 2.5. Phage Lysis

All 60 coliphages isolated from reference strain DSM 12242 were tested to lyse each of the clinical *E. coli* isolates, independent of the identified phenotypical resistance or of the identified resistance determinates. Only one phage was not able to lyse any of the clinical *E. coli*, and no phage was able to lyse all of the isolates.

Among the 57 *E. coli* isolates, none was lysed by all of the phages, but 15 were not lysed by any phage tested so far, and, therefore resistant to all phages. One *E. coli* isolate, which was most often affected by phages, was lysed by 53 out of the 60 phages isolated in this study (see Figure 1).

### 2.6. Restriction Analysis of Selected Phage DNA

A selected number (*n* = 29) of phages were further analyzed to confirm genetic difference between isolated phages. We cleaned (using DNase and RNase for digesting extracellular DNA and RNA), and precipitated lysates, isolated, and digested phage DNA—grown on reference strain DSM 12242. The restricted DNA of these phages resulted in six different restriction profiles (RP). To restriction profile one (RP1) belonged seven phages, ten to restriction profile 2 (RP2) and one phage each to restriction profile 3 (RP3) and 4 (RP4); six phages to RP5 and four to RP6 (see Figure 2 and Table 2). Restriction profiles were not consistent with host lysis ability, or testing positive or negative for resistant gene fragments. For example RP1, with seven phages consists of one phage that lysed only one clinical *E. coli* isolate, one lysed two isolates, another one lysed three isolates, one lysed six isolates, one lysed nine isolates, one lysed 15 isolates and another one lysed 20 *E. coli* isolates. In regards to resistant genes: one of these seven phages tested positive for resistant gene fragments *bla*_TEM_/*sulI*, one for *strA*, two for *strA*/*sulII*, and three for *bla*_TEM_/strA/*sulII* combination. RP2 harbored most phages. Two restriction profiles hold only one phage each (RP3 and RP4).

### 2.7. Difference in Lysis Properties of Resistant (Harboring Resistant Genes) versus Non-Resistant E. coli

To analyze if *E. coli* isolates with and without resistant genes differed in their resistance to phage lysis, we used the Fisher’s exact test. Isolates of *E. coli* harboring no resistant genes did not differ in their stability to phage lysis from those harboring one or more of the resistant genes (*p* = 0.361). Thus, all 15 *E. coli* resistant to lysis, by all phages, in part harbored resistant genes and in part did not. Phages harboring one or more resistant determinants did not differ in their ability to lyse *E. coli* isolates. The six phages that were able to transduce the *bla* gene into DSM 12242 all had high lytic capability.

## 3. Discussion

Disinfection regimes are becoming most important at clinics for small animals. Their main purpose is: (1) to avoid pathogens to spread from one patient to another; (2) to prevent persistence and transfer of pathogens at the clinics; (3) to hamper the colonization of medical staff and patients; and (4) to inhibit the transfer of pathogens with zoonotic potential to pet owners. Despite the challenge of preventing the transfer of pathogens (bacteria, virus, fungi), the transfer of antibiotic resistance with and between pathogens, is becoming increasingly important in these hot spots of antimicrobial use, and this applies to animal and human clinics likewise. Antibiotic resistance can be spread by a number of mechanisms, and will take place between pets and their owners, or between patients and medical staff [13,14]. Good disinfection regimes with appropriate agents (with virucidal components) will reduce this spread [12]. Nevertheless, interventions with colonized patients may bear a hazard not to be underestimated. Here we show that a dog with a chronic urinary tract infection harbors not only a community of *E. coli* bacteria with different antibiotic resistances against important antibiotics used to treat UTI, but hosts many different phages, some of which bear the ability to transduce antimicrobial resistance. Even though the dog had not been treated with antibiotics for over two years, multi-resistant *E. coli* cumulated in the urinary tract. None of the *E. coli* tested phenotypically sensitive to all eight antibiotics tested. All of the isolates were at least resistant to nalidixic acid, a quinolone mainly used for the treatment of urinary tract infections in both animals and humans [15,16]. The most common antibiotic resistant phenotype in the *E. coli* isolates was multi-resistant against a combination of ampicillin, streptomycin, sulfonamides, and nalidixic acid. Not only *E. coli* isolates harbored resistance genes (*bla*_TEM,_
*dfr, sulI*, *sulII, strA*) to the respective resistance phenotype, but also almost half of the phages isolated also tested positive for these very resistance genes. In the 60 phages isolated, we showed that 29 of them held parts of one or more of the resistance genes, as detected by PCR after cleaning the phage lysate with DNase to remove extra-viral DNA contamination. These genetic elements were the same in *E. coli* and the phages—e.g., the *sulI* gene only found in two *E. coli* isolates was also found in one phage, but the resistance element *sulII* found in 34 *E. coli* isolates was also found in 18 phages. Thus, the representation of these genes was evenly distributed in *E. coli* and their phages. This clearly shows a close exchange of resistance genes between *E. coli* and their phages, even though it is not clear if transduction could be seen as a most important mechanism for antibiotic resistance exchange between bacteria [13]. Direct resistance gene transduction with the extraction of phage lysates was not possible, but six phages were able to transduce ampicillin resistance after an infection of (grown on) a laboratory strain harboring puc19 without the transfer of the whole plasmid. Transductants were phenotypically resistant to ampicillin and harbored the *bla* gene of puc19, as detected by PCR.

By analyzing the DNA of 29 of the phages by digestion with restriction enzymes, six restriction profiles could be distinguished. Such data provided evidence for a multifaceted community of phages within the urinary tract. The phages were very different regarding their lysis of the *E. coli* isolates. No phage was able to lyse all of the *E. coli* isolates—this suggests that resistance against phages has been developed in some of the *E. coli*, from which 15 isolates showed resistance against all of the 60 phages.

Phage resistance in *E. coli* was independent of all of the antibiotic-resistant mechanisms detected. It was also independent of a multi-resistant phenotype, which could be an important result for the aspect of phage therapy against the UTI caused by *E. coli* [17].

Nowadays antibiotic resistance in UTI pathogens is quite common. Novel strategies have to be established to treat these infections. Phage therapy and the use of essential oils are most promising treatment options for UTI. Under these premises interaction with the pathogen, and the patient needs to be studied in every detail [18,19,20].

Although, in this study, we did not differentiate between pathogenic and non-pathogenic *E. coli*, we showed a broad variety of *E. coli* and coliphages that inhabit the urinary tract. The interaction between phages and their bacterial host has been shown to lead to a variety of reactions and counter-reactions—e.g., resistance against phage lysis and novel phage entry mechanisms [21].

Next to resistance development against phage lyses, phages can provoke phase variation in bacteria—this is a mechanism that produces phenotypic variation to allow a fast and reversible adaptation to stress full environments [22,23]. In *E. coli*, Type 1 fimbrial genes express the most common adhesins, which have a role in the virulence of urinary tract infections in both humans and animals [24]. These Type 1 fimbrial genes undergo phase variation in UTI in humans—this suggests their important role in virulence [25]. Considering that phages are able to induce phage variation in bacteria, such as *E. coli*, the interaction between phages and *E. coli* in case of urinary tract infections could be important for virulence, and possibly for antibiotic resistance development [26]. This needs to be further investigated.

## 4. Materials and Methods

### 4.1. Sample Collection and Preparation

We sampled the following locations: surgery table surface, examination table, and light and computer switches. These were sponged with sterile sponges and swabs. In total, 20 swabs and sponges were analyzed for *E. coli* phages and *E. coli*.

Additionally, at two different time points, ten days apart, urine from a dog visiting this clinic with a clinically diagnosed chronic urinary tract infection caused by *E. coli* was taken for routine health checks at the clinics using a sterile urinary catheter, and part of it (4 mL) was transported (under chilled conditions 3 °C) to the lab. Immediately after arrival, the urine was analyzed for *E. coli* and coliphages. The pet owner’s consent was obtained and ethical approval from the Ethics and Animal Welfare Committee of the University of Veterinary Medicine, Vienna, was received.

### 4.2. Isolation of E. coli and Bacteriophages

An aliquot of 100 µL of urine and 1:10 dilutions thereof were spread on Coli-ID agar plates (bioMérieux, Marcy l’Etoile, France) and incubated under aerobic conditions at 37 °C for 24 h. A total number of 57 single colonies were randomly isolated and sub-cultured on plate count agar for further analyses and for each isolate a separate stock in Luria broth (Merck, Darmstadt, Germany) with 20% glycerin (Sigma-Aldrich, St. Louis, MO, USA) was prepared and stored at −80 °C for further use. All isolates were typed biochemically as *E. coli* using the API 20E test system (bioMérieux, Marcy l’Etoile, France).

For bacteriophage isolation, samples (urine and liquids extracted from sponges and swabs) were filtered through a 0.22 µm filter. Coliphages were isolated according ISO 107006-2:2000 as previously described [4,27]. Single phage plaques (*n* = 60) were isolated and cultured as previously reported [28]. Lysates were filtered through a 0.22 µm filter to remove bacterial cell debris and stored at 4 °C for further analysis.

### 4.3. Antimicrobial Resistance Determination in E. coli Isolates

*E. coli* isolates were subjected to antimicrobial resistance testing using the Disc Diffusion testing method, according to routine laboratory testing following Clinical and Laboratory Standards Institute (CSLI) guidelines and described clinical breakpoints [29]. If no clinical breakpoints were defined, European Committee on Antimicrobial Susceptibility Testing (EUCAST) epidemiological break points [30] were used for resistance definition. *E. coli* were tested for ampicillin (AM10), chloramphenicol (C30), kanamycin (K30), nalidixic acid (NA30), streptomycin (S10), sulfonamides (S3 300), tetracycline (TE30), and trimethoprim (W5) resistance using *E. coli* ATCC 25922 as a non-resistant reference strain (Oxoid™ Thermo Fisher Scientific, Wesel, Germany). Genetic confirmation of resistance genes primers and PCR conditions used are stated in Table 3.

### 4.4. Antimicrobial Resistance Gene Detection in Phage Lysates

Phage DNA was isolated as described under 4.7. An aliquot of 50 ng DNA was used to perform PCR reaction according to Table 3.

### 4.5. Transduction of Resistance Genes

Resistance genes were transduced as previously described by us [4]. Briefly lysates with 10^4^, 10^5^ and 10^6^ plaque forming units (pfu)/mL (10 µL) positive for a certain resistance gene by PCR were incubated with *E. coli* host strain (DSM 12242) (100 µL o/n culture) for 30 min at 37 °C. After adding 2 mL modified Scholtens’ broth (MSB) the mixture was incubated under shaking for 2 h at 30 °C. After centrifugation at 8000× *g* for 3 min the pellet was re-suspended in 100 µL MSB and plated on Mueller Hinton Agar plates, with appropriate antibiotics for 48 h at 37 °C under aerobic conditions.

All 60 lysates were used to infect DH5α laboratory strain harboring puc19 at a concentration of 10^4^ pfu/mL. A fresh lysate was prepared of each of the sixty plates. The lysates were cleaned using a filtration through a 0.22 µm filter and a DNase/RNase digestion as described under 4.7 to eliminate bacteria and extra-viral DNA and RNA. These lysate preparations were used to transduce DSM 12242 as described above.

### 4.6. Determining Phage Lysis Profile of E. coli Isolates and Lysis Property of Phages

A spot screening assay was used for determining the lysis properties, as previously described [4]. In brief, 100 µL of an o/n culture of the host *E. coli* isolate was mixed with 2.5 mL preheated (42 °C) modified Scholtens’ soft agar and spread on modified Scholtens’ agar (MSA) plates. After solidification 5 µL drops of each obtained lysate with 10^5^ pfu/mL were spotted in a geometric pattern on the plate to leave enough space between the drops to form a suitable sized plaque. All *E. coli* isolates were tested for phage lysis with all the phages isolated in this study. After an incubation of 24 h at 30 °C formed plaques were judged as a positive or negative lysis reaction. This test was performed in duplicate for each phage and each *E. coli* isolate.

### 4.7. Phage DNA Isolation and Restriction Profile

Lysates were concentrated using precipitation by centrifugation with polyethylene glycol (PEG) and chloroform extraction as previously described [36]. In brief: same volumes of phage lysate and 20% PEG-8000/2.5 M NaCl solution were added and incubated on ice for 1 h. After a centrifugation step (10,000× *g*-20 min at 4 °C) the pellet was re-suspended in 100–500 µL SM-buffer (100 mM NaCl, 10 mM MgSO_4_, 50 mM TrisHCl, pH 7.5). Concentrated phage lysate was subjected to DNase (8 U/mL) and RNase (100 µg/mL) treatment before DNA extraction to eliminate contamination with extracellular DNA and RNA at 37 °C for 30 min. Phage DNA was prepared from a concentrated lysate (10^10^ pfu/mL). Restriction was performed with 500 ng phage DNA using 20 U PvuII-HF^®^ (20,000 units/mL in 1X CutSmart^®^ Buffer NEB #R3151, New England Biolabs) as recommended by the producer.

### 4.8. Statistical Analysis

To analyze if *E. coli* isolates with (i.e., one or more genes) and without resistance genes differed in their resistance to phage lysis (i.e., no lysis by any phage vs. lysis by at least one phage), we used Fisher’s exact test. The test was used since the combination presence of at least one resistance gene and no lysis contained merely 4 isolates. The same test was used to study if the ratio of *sulI* to *sulII* genes in *E. coli* would differ from that in coliphages.

## 5. Conclusions

Health care facilities are hot spots for antimicrobial resistant microorganisms. Transfer between patients, health care professionals, and accompanying persons (e.g., pet owners in small animal clinics) has to be eliminated by good hygiene practices, which demand not only effective disinfection against bacteria, but the use of virucidal compounds effective against viruses, including phages. Here, we show that patients can harbor *E. coli* and phage, able to transduce antimicrobial resistance. In chronic urinary tract infections from dogs, antimicrobial resistant bacteria can stay, and resistance genes can exchange with phages, at least, as long as two years after the last antibiotic treatment. These phages are variable in lytic and transduction ability. Some *E. coli* have developed diverse resistances to most, if not all, phages. This is important to note for possible forthcoming phage therapy options.

## Figures and Tables

**Figure 1 antibiotics-09-00652-f001:**
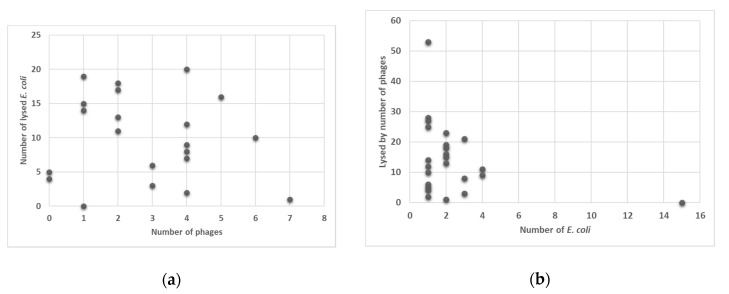
Ability of phages to lyse none, one, or more *E. coli* (*y*-axis) (**a**) and susceptibility of the *E. coli* isolates to be lysed by none, one, or more of the 60 phages (*y*-axis) (**b**).

**Figure 2 antibiotics-09-00652-f002:**
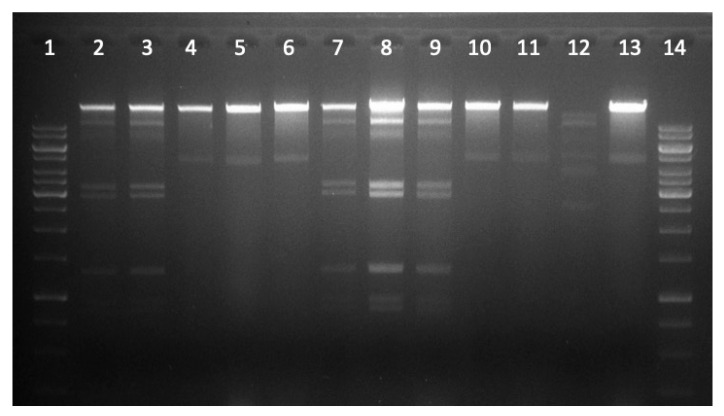
Restriction profile of phage DNA using *PvuII*. This gel documents 12 restricted phages in which three restriction profiles could be distinguished. Slot 1 and 14 molecular weight marker, slot 2, 3, 7, 8, 9 restriction profile 1 (RP1), slot 4, 5, 6, 10, 11, 13 restriction profile 2 (RP2), slot 12 restriction profile 3 (RP3).

**Table 1 antibiotics-09-00652-t001:** Phenotypical resistance and resistance gene combination in clinical *E. coli* isolates from a chronic urinary tract infection (UTI) of a dog.

Resistance	AM ^R^	NA ^R^	S ^R^	S3 ^R^	W ^R^
*bla* _TEM_	39	39	34	34	1
*dfr*	1	1	0	0	1
*sulI*	2	2	2	2	0
*sulII*	34	34	34	34	0
*strA*	34	34	34	34	0

AM ^R^ NA ^R^ S ^R^ S3 ^R^ W ^R^ phenotypical resistance to ampicillin, nalidixic acid, streptomycin, sulfonamides, trimethoprim, respectively.

**Table 2 antibiotics-09-00652-t002:** Restriction profiles of 29 phages after digestion of phage DNA with *PvuII.*

RP1	RP2	RP3	RP4	RP5	RP6 ^1^
PNap5	PNap7	PNap50	PNap22	PNap13	PNap16
PNap6	PNap11			PNap15	PNap17
PNap29	PNap12			PNap44	PNap20
PNap31	PNap19			PNap45	PNap56
PNap33	PNap25			PNap46	
PNap47	PNap26			PNap54	
PNap49	PNap28				
	PNap38				
	PNap43				
	PNap53				

^1^ undigested DNA.

**Table 3 antibiotics-09-00652-t003:** Primers and conditions for PCR of tested resistance genes.

Primer	Sequence	Target Gene	Amplicon Size	Annealing (°C)	Reference
bla_TEM-1_-f	5′-cagcggtaagatccttgaga-3′	*bla* _TEM_	643	55	[31]
bla_TEM-1_-r	5′-actccccgtcgtgtagataa-3′				
bla_CMY_-f	5′-tggccgttgccgttatctac-3′	*bla* _CMY_	870	55	[31]
bla_CMY_-r	5′-cccgttttatgcacccatga-3′				
dfr1-f dfr1-r	5′-gtgaaactatcactaatgg-3′ 5′-ttaacccttttgccagattt-3′	*dfrA1, dfrA5, dfrA15, dfrA15b, dfrA16, dfrA16b*	474	55	[32]
sulI-f sulI-r	5′-tggtgacggtgttcggcattc3′ 5′-gcgaaggtttccgagaaggtg-3′	*sul*I	790	63	[33]
sulII-f sulII-r	5′-gcgctcaaggcagatggcatt-3′ 5′-gcgtttgataccggcacccgt-3′	*sul*II	293	60	[34]
strA-B-f strA-B-r	5′-ccaatcgcagatagaaggcaag-3′ 5′-atcaactggcaggaggaacagg-3′	*strA*	580	65	[35]

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
