# Peer review of "Antimicrobial Resistance in Escherichia coli and Resistance Genes in Coliphages from a Small Animal Clinic and in a Patient Dog with Chronic Urinary Tract Infection"

_antibiotics, 2020, doi:10.3390/antibiotics9100652_

Round 1
Reviewer 1 Report
Dear Authors
Analysis by working partitions:
1 - Highlights: they have been displayed correctly, there are no revisions to make
2 - Introduction: must be reformed in the content and in the writing of the general part review the syntax of the topic
3 - Materials and Methods clearly exposed and valid methods for the detection of resistance genes in E. coli
4- Discussion :deepen the fact of the scarcity of means, given the increasing antibiotic resistance against E. coli insert five lines (using the references that I indicate you will have to quote) on natural products active in urinary tract infections which could be a Valid alternative:
PMID: 32344551 ; PMID: 29987237 ; PMID: 31926578
5 - Check the bibliographic entries throughout the text, some of which are non-compliant, review some entries in the references and necessarily insert those referred to in point 4 for the purpose of acceptance by me.
6 - Review the English grammar and in particular the applied scientific English: in particular, verbal tenses and syntax in the discussion.
Author Response
Dear reviewer,
Thanks for your comments and suggestions.
We have edited the manuscript especially to your specific and general comments.
- The section „Introduction“ was revised. See track changes and underlayed yellow parts in the revised manuscript.
- The section „Discussion“ has been revised and a paragraph of novel UTI treatment options has been inserted.
- Biobliography has been revised in detail (see track changes)
- The manuscript has been edited by a native Englishman – see track changes
Best regards Friederike Hilbert and all co-authors
Reviewer 2 Report
The study plan and results of the manuscript were interesting. The manuscript fitting well with the scope of the Antibiotics Journal and indicated good scientific value. I would recommend this manuscript to be published after a minor revision for the following comments.
- Title, Page 1, line 2: Write E. coli in full form and correct the font case the word ‘resistance’
- Abstract, Page 1, Line 17: Correct ‘we isolates’
- Introduction, Page 1, Paragraph 1, Line 31: Briefly mention the various mechanisms of bacterial resistance development and transfer.
- Introduction, Page 1, Paragraph 2, Line 36: List the names of multidrug-resistance pathogens mentioned in the literature.
- Material and Methods, Section 4.1, Line 208: Correct ‘Sampling sites as are’
- Material and Methods, Section 4.1, Line 213: Correct ‘ml’ as mL’ and add a space between the number and unit.
- Material and Methods, Section 4.1, Line 215: Mention the ethical approval number.
- Throughout the manuscript check and correct the units (ml, µl, etc.) to an international standard format with a space between the number and unit.
- Material and Methods, Section 4.3, Line 234: Write the full form of CSLI at its first use. Also, check for other abbreviations at their first use.
- Material and Methods, Section 4.8: Correct the font of this paragraph.
Author Response
Dear reviewer 2
Thanks for your suggestions and comments on the manuscript.
We revised the manuscript according to your statements as follows:
- The title has been revised for font case correction and E. coli has been spelled in full.
- The abstract has been revised “we isolated”
- The "Introduction" section was revised: mechanisms of resistance development have been itemised (see track changes and yellow underlay).
- In the Introduction section most important multi-drug resistant pathogens have been itemised (yellow underlay)
- For the “material and methods” section we rephrased the “Sampling sites as are” to “We sampled the following locations…”
- We corrected all units to international standards and entered a space as suggested
- The analyses of left-over from clinical diagnostic material has to be announced to the Ethic and Animal Welfare Committee of the University of Veterinary Medicine, Vienna and the approval has to be given, but no approval number has to be assigned.
- CLSI and EUCAST have been spelled in full in the M&M section and the manuscript has been reviewed for other abbreviations.
- The font of 4.8. has been corrected
- The manuscript has been edited by a native Englishman – see track changes
Reviewer 3 Report
The paper title “Antimicrobial resistance in E. coli and Resistance Genes in Coliphages from a Small Animal Clinics and in a Patient Dog with Chronic Urinary Tract Infection” emphasizes the importance of applying a good hygiene practice in a veterinary clinic and antimicrobial resistance of E.coli and coliphages. The subject of the manuscript is interesting and essential to increase a knowledge of antimicrobial resistance of E.coli and coliphages isolated from the urinary tract of the patient dog. Before publication, the manuscript should be read from beginning to end by all authors and appropriate corrections should be made to harmonize and improve the text, recommended English correction.
Minor comments:
Line 16: “In none of the 20 samples of the surgery unit and diagnostic rooms coliphages were isolated.” In my opinion “E.coli” is missing
Line 17: Authors should check “we isolates” or “we isolate”
Line 18-19: not clear it is results or methodology
Line 23: change “harboring” to “harbouring”
Line 28: “raising”
Line 33: check to spell “predominate” or “predominant”
Line 37 and 52: abbreviation E.coli
Line 44: “relying”
Line 49; The aim of the study is not clear. The Authors should check in all manuscript coliphages and E.coli because like in Line 16 the Authors forgot about E.coli
Line 58: in all text, the Authors use “coliphages” here I have “E. coli phages”
Line 72: remove “a period of”
Line 74: remove “to” PCR and “a” detection
Line 89: “harboured”
Line 94: “were” not “where found by PCR”
Line 137: Figure 2. The Authors have to correct the numbers
Line 154-156: The Authors should think about change this sentence maybe the numbers change for the words: first, second….etc.
Line 164: “harbours”
Line 172: “harboured” Authors should check in all text
Line 250: Change the sentence “Transduction of resistance genes was performed as previously described in [12].”
Line 312: The “References” should be written according to “Instructions for the Authors”
Author Response
Dear reviewer 3
Thanks for your suggestions and comments on the manuscript
All authors have read and revised the adjusted manuscript according to your statements.
The manuscript has been edited by a native Englishman – see track changes
Line 16: E. coli has been added. The sentence states now: In none of the 20 samples of the surgery unit and diagnostic rooms, neither the coliphages nor E. coli were isolated.
“We isolates” was changed to “we isolated”
We rephrased line 18-19: All the E. coli isolates were determined as resistant against nalidixic acid, 47 against ampicillin, 34 against sulfonamides and 33 against streptomycin.
The verb haboring was changed throughout the document to “harbouring
Sentence in line 28 has been rephrased according to revised English language.
Line 33 “predominate” now line 34 has been changed to “predominant”
Throughout the manuscript but the title, first time in the abstract and key words: Escherichia coli has been abbrivated to E. coli
Line 44 now line 98 “relaying” has been exchanged to “relying”
Line 49 now line 103 and throughout the document: E. coli was added.
Line 58 now line 112: E. coli phages have been changed to coliphages
Line 72 now line 127: “a period of” has been removed
Line 74 now line 129: “to” and “a” has been removed
Line 89 now line 197 and throughout the document “harbored” has been changed to “harboured”
Line 94 now line 202: “where” was changed to “were”
Figure 2: numbers have been adjusted
Line 154-156: English editing has been done. The sentence says now: Their main purpose is: 1. To avoid pathogens to spread from one patient to another; 2. to prevent persistence and transfer of pathogens at the clinics; 3. to hamper the colonization of medical staff and patients; and 4. to inhibit the transfer of pathogens with zoonotic potential to pet owners.
Line 250 now line 606: reads now: Resistance genes were transduced as previously described by us (4).
Line 312 now line 671: References were edited. See track changes.
Round 2
Reviewer 1 Report
The suggested corrections were performed successfully.